# The Influence of Methods for Cardiac Output Determination on the Diagnosis of Precapillary Pulmonary Hypertension: A Mathematical Model

**DOI:** 10.3390/jcm12020410

**Published:** 2023-01-04

**Authors:** Léon Genecand, Gaëtan Simian, Roberto Desponds, Julie Wacker, Silvia Ulrich, Benoit Lechartier, Jean-Marc Fellrath, Olivier Sitbon, Maurice Beghetti, Frédéric Lador

**Affiliations:** 1Division of Pulmonary Medicine, Department of Medicine, Geneva University Hospitals, 1211 Geneva, Switzerland; 2Faculty of Medicine, University of Geneva, 1211 Geneva, Switzerland; 3Pulmonary Hypertension Program, Geneva University Hospitals, 1211 Geneva, Switzerland; 4Faculty of Mathematics, University of Geneva, 1205 Geneva, Switzerland; 5Pediatric Cardiology Unit, Department of Pediatrics, Gynecology, and Obstetrics, Geneva University Hospitals, 1211 Geneva, Switzerland; 6Department of Pulmonology, University of Zurich, University Hospital of Zurich, 8091 Zurich, Switzerland; 7Division of Respiratory Medicine, Lausanne University Hospital, 1011 Lausanne, Switzerland; 8Pulmonary Medicine Unit, RHNe Réseau Hospitalier Neuchâtelois, Pourtalès and la Chaud-de-Fonds Hospitals, 2000 Neuchatel, Switzerland; 9Université Paris-Saclay, INSERM UMR_S999, Assistance Publique–Hôpitaux de Paris, Service de Pneumologie et Soins Intensifs Respiratoires, Centre de Reference de l’Hypertension Pulmonaire, Hôpital de Bicêtre, 94270 Le Kremlin Bicêtre, France

**Keywords:** pulmonary hypertension, cardiac output, thermodilution, direct Fick, mathematical model

## Abstract

Background: precapillary pulmonary hypertension (PH, PcPH) is now defined as a mean pulmonary artery pressure (mPAP) > 20 mmHg, a pulmonary artery wedge pressure (PAWP) ≤ 15 mmHg and a pulmonary vascular resistance (PVR) > 2 WU. For PVR calculation, the measurement of cardiac output (CO) is necessary. It is generally measured using thermodilution. However, recent data showed that the agreement with direct Fick method, historically the gold standard, is less than previously reported. We aimed to create a mathematical model that calculated the probability of being classified differently (PcPH or unclassified PH) if CO measured by direct Fick was used instead of thermodilution for any individual patients with a mPAP > 20 mmHg and a PAWP ≤ 15 mmHg. Methods: The model is based on Bland and Altman analysis with a normally distributed difference of cardiac output, fixed 1.96 standard deviation of bias, bias and physiological cardiac output limits. Results: Following a literature review of the studies comparing CO measured with direct Fick and thermodilution, we fixed the 1.96 standard deviation of bias at 2 L/min, bias at 0 L/min and physiological resting CO limits between 1.3 L/min and 10.2 L/min. Conclusions: This model can help the clinician to evaluate the potential benefit of measuring CO using direct Fick during the diagnostic work-up and its utility in confirming or ruling out a diagnosis of PcPH in any given patient with a mPAP > 20 mmHg and a PAWP ≤ 15 mmHg.

## 1. Introduction

Pulmonary hypertension (PH) is now defined as a rise of mean pulmonary artery pressure (mPAP) > 20 mmHg measured during right heart catheterisation (RHC) [1]. The addition of a pulmonary vascular resistance (PVR) > 2 WU and a pulmonary artery wedge pressure (PAWP) ≤ 15 mmHg defines all types of precapillary pulmonary hypertension (PcPH) [1]. The PVR is useful to differentiate between an increase in mPAP caused by a high cardiac output (CO) or a high PAWP and pulmonary vascular disease with increased PVR leading to an increase in mPAP for a given PAWP and CO. PVR is calculated by dividing the difference between the mPAP and the PAWP by the CO (PVR = (mPAP − PAWP)/CO). Precise measurements of mPAP, PAWP and CO are thus essential for PVR calculation.

The gold standard for estimating CO is the direct Fick (DF) method. CO measured using DF (CO_DF_) is calculated by dividing the oxygen consumption (V′O_2_) by the difference between the arterial oxygen content (CaO_2_) and mixed venous oxygen content (Cv′O_2_) (CO_DF_ = V′O_2_/(CaO_2_ − Cv′O_2_)). While CO_DF_ is acknowledged to be the most accurate and precise method to determine CO, its measurement is cumbersome due to the measurement of V′O_2_ during RHC and the necessity of arterial and mixed venous blood drawl and analysis, which challenges repetitive assessment in transient haemodynamic situations such as during exercise.

CO measurement using thermodilution (TD, CO_TD_) widely replaced CO_DF_ in the field of PH based on the results of one study comparing CO_TD_ against CO_DF_ and demonstrating acceptable agreement between the two methods in 35 PcPH patients [2]. Recent studies, however, did not confirm the previous results and demonstrated a poorer agreement between CO_TD_ and CO_DF_ using Bland and Altman (BA) analysis for patients with PcPH [3,4], or with a suspicion of PH [5].

As PVR calculation is dependent on the estimation of CO, a difference in estimated CO values using the different methods could result in a different diagnosis and classification of PH. Indeed, according to the currently proposed haemodynamic definition of PH, a patient with a mPAP > 20 mmHg, a PAWP ≤ 15 mmHg and PVR of >2 WU is classified as having PcPH, while a patient with a mPAP > 20 mmHg, a PAWP ≤ 15 mmHg but a PVR of ≤2 WU is classified as having unclassified PH [1].

The influence of using CO_TD_ value instead of CO_DF_ on the diagnosis of PcPH is unknown. Thus, we aimed to create a mathematical model which estimates the probability of a patient with a mPAP > 20 mmHg and a PAWP ≤ 15 mmHg being classified differently (PcPH or unclassified PH) if CO_DF_ was used instead of CO_TD_.

## 2. Materials and Methods

### 2.1. Study Population

This study is based on a mathematical model and a literature review of previous published studies comparing the agreement between CO_DF_ and CO_TD_ in PcPH patients.

### 2.2. Mathematical Model

In order to create this mathematical model, it was necessary to estimate the agreement between CO_TD_ and CO_DF_ using BA statistics. A BA graph is one statistical method used for evaluating two methods comparing the same variable [6]. The BA graph plots the differences of the variable measured using the two different methods against their mean. This graph mainly shows the mean difference between the two methods (i.e., bias) and the 1.96 standard deviation (SD) of the bias also called the “limits of agreement” (LoA), which contains 95% of the differences when the distribution of the differences follows a normal distribution. The 1.96 SD of bias estimates the agreement between the two methods: the bigger the 1.96 SD of bias, the poorer the agreement between the two methods. When two methods comparing the same variable are evaluated, the differences between their measurements often follow a normal distribution [6]. This occurs as, by comparing a variable obtained in a same person, the individual variation is eliminated and the comparison of the two methods depends mainly on measurement errors [6].

This mathematical model is based on the following elements: a normal distribution of the CO differences as well as fixed bias, 1.96 SD of bias and physiological CO limits at rest.

The creation of this model followed a three-step approach. Firstly, a literature review of studies comparing CO_TD_ and CO_DF_ in PcPH was performed to evaluate the most appropriate bias, 1.96 SD of bias, physiological CO limits at rest and to confirm that the CO differences between CO_TD_ and CO_DF_ probably follows a normal distribution in this population. Second, we created two sets of fictive CO values, with their differences following a normal distribution based on the most probable bias, 1.96 SD of bias and physiological CO limits at rest in PcPH. Lastly, we created the model allowing an assessment of the probability for any individual to have a different diagnosis using CO_TD_ or CO_DF_.

#### 2.2.1. Comparison between CO_TD_ and CO_DF_ at Rest in PcPH Patients

We updated our recent literature review concerning the methods of cardiac output measurement in PcPH patients but with a focus only on CO_TD_ and CO_DF_ [7]. We included articles published until May 2022. The following filters were applied to PubMed: (“thermodilution” OR “Direct Fick”) AND “pulmonary hypertension” and the following filter on web of science: Set 1: TI = (Pulmonary Hypertension), set 2: TI = ((thermodilution) OR (direct Fick)). We then associated “Set 1 AND Set 2”. The PubMed search yielded 146 articles. 132 articles were excluded after reading the abstract or based on their titles. 14 were selected for full text reading. Of those 14, 8 studies did not compare TD and DF and 3 studies used TD and DF but did not assess the agreement between these methods [8,9,10]. These articles were subsequently excluded. Three articles comparing CO_TD_ and CO_DF_ in PcPH using BA analysis were finally included and are summarised in Table 1 [2,3,4].

#### 2.2.2. Creation of a Set of Data with a Normal Distribution of CO Differences within a Plausible CO Range

We assumed that the difference between CO_DF_ and CO_TD_ followed the normal distribution of:CO_TD_ − CO_DF_ = X~N (μ,σ)(1)

On the BA graph, μ corresponds to the bias and σ corresponds to the SD of the CO differences. For a random variable X following a normal distribution N (μ,σ), X will lie in the interval [μ − 1.96 σ, μ + 1.96 σ] with a probability of 95% [6]. By fixing the 1.96 SD of the bias, the SD could be calculated.

A large amount of random data was then generated following a normal distribution of CO differences on a BA graph using Python (version 3.10.4) with fixed bias, 1.96 SD of CO differences and limits of physiological CO at rest. The normal distribution of the generated data was then verified using a histogram.

#### 2.2.3. Diagnostic Disagreement Calculation with Limits of Plausible CO Values

A model was then created to calculate the individual probability of being classified differently (PcPH or unclassified PH) if CO_DF_ was used instead of CO_TD_ based on the patient’s haemodynamic characteristics (mPAP, PAWP and CO_TD_).

We defined diagnostic disagreement (DgDis) as the probability for a given patient to be misdiagnosed using the method CO_TD_ compared to the method CO_DF_. There are two possible scenarios. Either a diagnosis of PcPH would be made using CO_TD_ but not using CO_DF_, or a diagnosis of PcPH would be made using CO_DF_ but not using CO_TD_. The first scenario was defined as DgDis+, meaning a false positive diagnosis and the second scenario as DgDis−, which corresponded to a false negative diagnosis.

We required both CO_TD_ and CO_DF_ to be in a defined interval [a, b] and the difference between CO_TD_ and CO_DF_ to follow a normal distribution. In this context, the mean value of CO_TD_ and CO_DF_ (i.e., (CO_TD_ + CO_DF_)/2) obligatory lies in the same interval [a, b] as both individual CO measurements (CO_TD_ and CO_DF_).

To implement computation with these conditions, we used conditional probabilities. For two events A and B, the probability of A conditioned to B is the probability that the event A happens, knowing that the event B has already occurred. This is defined through the formula:P(A|B) = P(A∩B)/P(B)(2)

In our case, we knew that CO_DF_ lies within the defined interval of physiological CO [a, b], (event “B”). Event “A” is the probability that a DgDis exists, meaning that the use of CO_DF_ leads to another diagnosis than the use of CO_TD_.

Thus, we needed to compute P(A∩B) and P(B), which are given by the following steps.

If the difference between CO_TD_ and CO_DF_ follows a normal distribution:CO_TD_ − CO_DF_ = X~N (μ,σ), (3)
then CO_DF_ = CO_TD_ − X(4)

The event B corresponds to CO_DF_ lying in [a, b], this means that CO_TD_ − X ≥ a and CO_TD_ − X ≤ b, which is equivalent to X ≤ CO_TD_ − a and X ≥ CO_TD_ − b.

Thus, the following event B corresponds to the inequalities (CO_TD_ − b ≤ X ≤ CO_TD_ − a).

For a patient having PcPH defined by CO_TD_ measurement, the event DgDis+ corresponds to the transpulmonary gradient (TPG), which is the difference between mPAP and PAWP, divided by CO_DF_, equal to or less than 2: TPG/CO_DF_ ≤ 2. This corresponds to the absence of PcPH using CO_DF_.

This is equivalent to TPG/(CO_TD_ − X) ≤ 2. So DgDis+ corresponds to CO_TD_ − TPG/2 ≥ X.

Analogous reasoning shows that DgDis− corresponds to CO_TD_ − TPG/2 ≤ X.

In summary, the probability needed to compute the above conditional probabilities are given by:P(DgDis+ ∩ B) = P(CO_TD_ − b ≤ X ≤ min(CO_TD_ − TPG/2, CO_TD_ − a))(5)
P(DgDis− ∩ B) = P(max(CO_TD_ − TPG/2, CO_TD_ − b) ≤ X ≤ CO_TD_ − a)(6)
P(B) = P (CO_TD_ − b ≤ X ≤ CO_TD_ − a),(7)
where the TPG is equal to mPAP − PAWP.

## 3. Results

Table 1 summarises the studies comparing CO_TD_ and CO_DF_ in PcPH [2,3,4]. In total, 113 patients had pulmonary arterial hypertension (PAH) (group 1) patients and 21 patients had chronic thromboembolic PH (CTEPH). The bias varied between 0.01 and 0.45 L/min. The 1.96 SD of bias varied between 1.1 L/min and 2.48 L/min. The limits of mean resting CO extracted visually from the BA graph were between 1.6 L/min (lowest mean CO observed from Hoeper et al.) and 8.5 L/min (highest CO observed by Duknic et al.). None of the authors demonstrated a proportional bias (i.e., increasing or decreasing CO difference with CO mean). This supports the hypothesis of a normal distribution of the CO differences. Based on these results, we arbitrarily chose to construct our model with a bias of 0 L/min, 1.96 SD of 2 L/min and limits of physiological resting CO between 1.3 L/min and 10.2 L/min (observed mean CO limits with a subjective 20% safety margins).

**Table 1 jcm-12-00410-t001:** Studies comparing CO_TD_ and CO_DF_ in PcPH.

	Patients	mPAP	Mean CO_DF_	Mean CO_TD_	Bias	1.96 SD of Bias
Hoeper et al. [2]	35 (31 PAH, 4 CTEPH)	56 ± 12	3.7 ± 1.2	3.7 ± 1.3	0.01	1.10
Khirfan et al. * [4]	75 PAH patients	48 ± 14	4.6 ± 1.5	4.6 ± 1.6	0.02	2.02
Duknic et al. [3]	24 (7 PAH 17 CTEPH)	37 ± 11	5.9 ± 1.5	5.5 ± 1.2	0.45	2.48

* Data are given in cardiac index in the original article. CO data are given based on oral communication of the corresponding author Dr. Tonelli. CO_TD_: Cardiac output measured using TD. CO_DF_: Cardiac output measured using direct Fick; CTEPH: chronic thrombo-embolic pulmonary hypertension; mPAP: mean pulmonary artery pressure; PAH: pulmonary arterial hypertension; SD: standard deviation.

Figure 1 shows a BA graph of two randomly created CO sets with the differences of CO following a normal distribution. The data were randomly created using Python based on a bias of 0 L/min, 1.96 SD of 2 L/min and resting CO limits between 1.3 L/min and 10.2 L/min. The histogram showing the distribution of the CO differences exposed in Figure 1, confirms the normal distribution of the data as shown in Figure 2.

Table 2 and Table 3 give examples of fictitious patients with corresponding probabilities of DgDis− and DgDis+, respectively using the mathematical model. Figure 3 plots the TPG and the CO. The diagnosis cut-off isoPVR line (2 WU) and DgDis lines corresponding to a probability of 10% and 20% are pictured. The isoPVR lines of 1 WU and 3 WU are shown in Figure 4.

## 4. Discussion

In this study, we presented the first mathematical model that allows the calculation of the probability of DgDis if CO_DF_ was used rather than CO_TD_ for a given individual with a mPAP > 20 mmHg and a PAWP ≤ 15 mmHg.

This work introduces the notion of doubt related to CO determination in the catheterisation laboratory, which results in a systematic questioning concerning the reliability of CO measurement using TD and its influence on the diagnosis of PcPH.

The model can be used both for patients being categorised as having either PcPH (mPAP > 20 mmHg, PAWP ≤ 15 mmHg and PVR > 2 WU) or as having unclassified PH (mPAP > 20 mmHg, PAWP ≤ 15 mmHg and PVR ≤ 2 WU) using CO_TD_ according to the new PH haemodynamic definition [1].

The model’s findings highlighted two intuitive rules: the DgDis is high if the patient lies close to a PVR of 2 WU and the DgDis is higher for a given PVR for lower CO or a lower TPG. In other terms, since the PVR is a ratio between the TPG and the CO, the smaller the TPG and CO are in absolute terms, the greater is the impact of a given variability on the calculation of the PVR and thus the diagnosis disagreement.

When considering these rules, one should be particularly aware of patients with a PVR close to 2 WU and with low CO. For example, a patient diagnosed with unclassified PH using TD with a mPAP of 21 mmHg a PAWP of 14 mmHg and a CO_TD_ of 4 L/min (PVR = 1.75 WU) has a 29.4% estimated probability of being diagnosed as having PcPH using CO_DF_ (Table 2, row 2).

On the other hand, it is possible to be confident in the haemodynamic diagnosis using CO_TD_ in a patient with very elevated or very low PVR. For example, a patient diagnosed with PcPH using TD based on a mPAP of 21 mmHg, a PAWP of 11 mmHg and a CO_TD_ of 3 L/min (PVR = 3.3 WU), has a 3% of chance of being diagnosed as having unclassified PH using DF, which is very reassuring (Table 3, row 8).

Based on the model; we propose three distinct situations:For a DgDis < 10%, the clinician can be quite confident of a diagnosis using CO_TD_ and measurement of CO_DF_ is not systematically required.For a DgDis between 10 and 20% the clinician should consider the use of CO_DF_.For a DgDis > 20% the clinician should use CO_DF_ because the probability that the patient would be classified differently using CO_DF_ is high.

In our model, default settings including no systematic bias, a 1.96 SD of bias of 2 L/min and resting CO in between 1.3 L/min and 10.2 L/min were used. We considered this was the most appropriate subjective default settings considering the actual literature. However, it must be acknowledged that there a paucity of data, with only three studies cumulating a total of 134 patients with 21 CTEPH and 113 PAH patients. As shown in Table 1, the exact bias and 1.96 SD of bias between TD and DF in the context of PcPH is unknown and there is variability in the published studies. Possible explanations for the observed variability include: different severity and characteristics of PcPH patients, with lower mean CO in the study from Hoeper et al. [2], and differences in the protocol of CO measurement using either TD or DF (e.g., average of two to three measurements for TD for Duknic et al. [3], five measurements with deletion of highest and lowest values for TD for Hoeper et al. [2]).

We assumed a normal distribution of the difference, which implies that there is no proportional bias when comparing CO_TD_ and CO_DF_ (i.e., the differences systematically increase or decrease with increasing CO). Reassuringly a proportional bias was not observed in the included studies [2,3,4]. This is incorrect during exercise with high CO where a proportional bias is often observed using TD and the assumption of a normal distribution between the differences of CO is therefore lost [3,11]. Consequently, the presented mathematical model should not be used in this context.

The resting limits of CO were based on real life data of CO_TD_ and CO_DF_ in PcPH using the lowest mean CO (Hoeper et al. 1.6 L/min) and the highest mean CO (Duknic et al. 8.5 L/min) and adding safety margin of 20% yielding CO limits between 1.3 L/min to 10.2 L/min [2,3]. The consequence of fixing boundaries of possible CO is the creation of a distortion in the linear DgDis line when approaching the limit of the model as seen in Figure 3 and Figure 4. We considered this was the best mathematical representation of the real life where an aberrant CO measurement (e.g., 1 L/min) would probably be rejected by the clinician.

The main limitations of the model are the unknown exact 1.96 SD of bias in PcPH patients and the assumptions that the CO differences followed a normal distribution in all settings. More data are needed to understand the exact 1.96 SD of bias and to assure that CO differences follow a normal distribution in a wide variety of PcPH patients and haemodynamic severities. In addition, patients with the mPAP of 21, 22, 23 or 24 mmHg were not included in the presented studies and the 1.96 SD of bias might be different for patients with low mPAP. Published studies did not, however, find differences in agreement with increasing tricuspid regurgitation, which is linked to an increase mPAP [2,4]. Bearing this in mind, we hypothesized that the agreement between TD and DF would not significantly differ for patients with a lower mPAP.

The strengths of the model are an individualised approach for a given patient and the highlighting of the influence of CO measurements agreement on the diagnosis of PcPH. Additionally, defaults settings of the model could be easily adapted if new evidence on the agreement of TD and DF is published.

Although DF remains the gold standard in measuring CO, its systematic use is difficult to implement due to its complexity and its time-consuming nature. In practice its use is often restricted to situations where TD is known to give false results as for patients with intracardiac shunts. This mathematical model can help to identify patients who would benefit most from DF. Either an incorrect diagnosis of PcPH or an incorrect diagnosis of the absence of PcPH could have serious implications. Our model could aid clinicians to adopt a personalised strategy when evaluating the potential added value of the use of DF and may lead to a more accurate diagnosis.

Cardiac output measurement using indirect Fick is frequently used in clinic. However, the agreement between IF and TD for patients with PH or suspicion of PH is poor with LoA ranging between 1.8 L/min to 4.1 L/min and bias ranging between 0.4 L/min and −0.8 L/min [5,12,13,14]. Comparisons of indirect Fick with DF in PcPH are lacking [7]. For this reason IF is not recommended anymore in the international guidelines because it is considered less reliable than TD and DF [1]. Other non-invasive methods have been studied in PcPH including inert gas rebreathing, bioimpendance, bioreactance, pulse wave analysis, cardiac magnetic resonance imaging and echocardiography [7]. The most promising method might be cardiac magnetic resonance imaging that showed excellent agreement when compared to TD or DF with LoA close to 1 L/min [15,16]. Since the characteristics of the presented mathematical model were based on the agreement between TD and DF, it cannot be readily used for other methods, such as indirect Fick or cardiac magnetic resonance, that have different agreement with the gold standard DF. However, the mathematical model could be modified to study the influence on diagnosis disagreement of any given CO methods assuming that the LoA and bias are known and that the CO differences of the studied methods have a normal distribution.

The principle of this mathematical model could also be adapted to study the influence on the diagnosis of PcPH of different parameters of the PVR equation. For example, the mPAP can be estimated using cardiac magnetic resonance [17]. The mathematical model could be modified to study the diagnosis influence of the mPAP estimated using cardiac magnetic resonance compared to the mPAP measured during RHC.

## 5. Conclusions

Using the new haemodynamic definition of PcPH, we created a model that estimated the probability of a different diagnosis (PcPH or unclassified PH) when using CO_TD_ instead of CO_DF_ based on the patient individual haemodynamic characteristics. This model can help the clinician to evaluate the potential benefit of measuring CO_DF_ during the diagnostic work-up of PcPH in any given patient with a mPAP > 20 mmHg and a PAWP ≤ 15 mmHg. Moreover, it emphasises the need to question the reliability of TD over DF for the diagnosis of PcPH for every patient. Such an approach could increase the diagnosis accuracy, and thus be beneficial to patients.

## Figures and Tables

**Figure 1 jcm-12-00410-f001:**
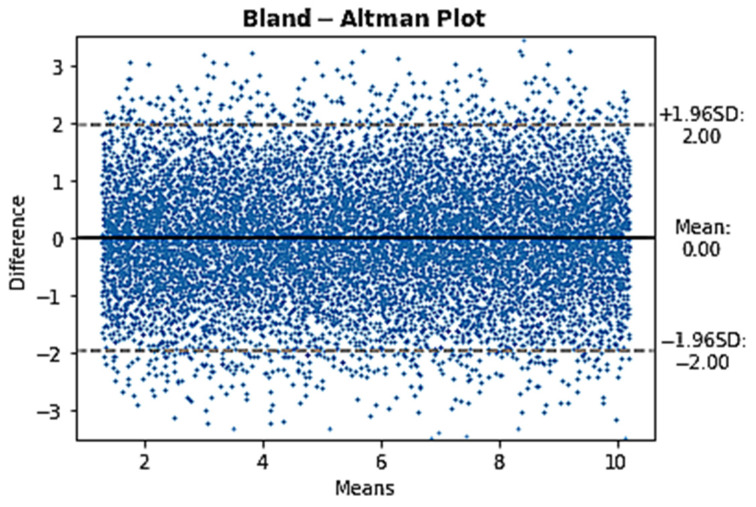
Bland and Altman graph showing a normal distribution of CO differences based on the model’s characteristics with fictive data. Randomly generated data using Python following the model’s characteristics (1.96 SD of 2 L/min, bias of 0 L/min and limits of CO of 1.3 and 10.2 L/min) and a normal distribution of CO differences. SD: standard deviation.

**Figure 2 jcm-12-00410-f002:**
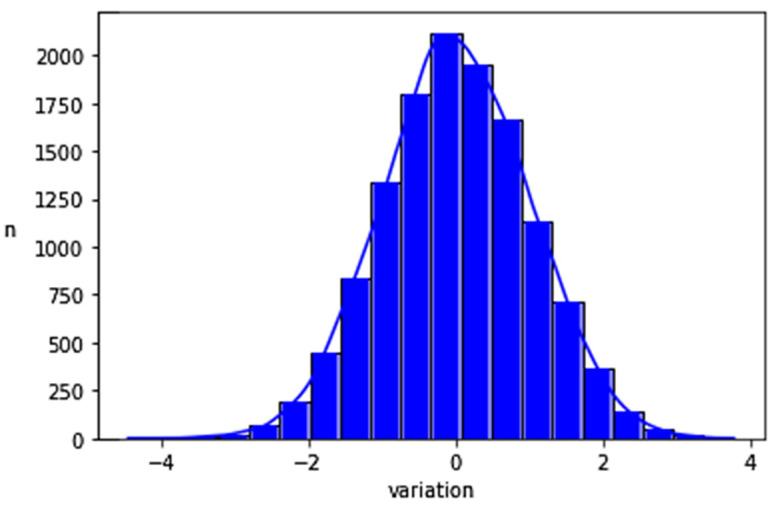
Histogram of the CO differences following a normal distribution.

**Figure 3 jcm-12-00410-f003:**
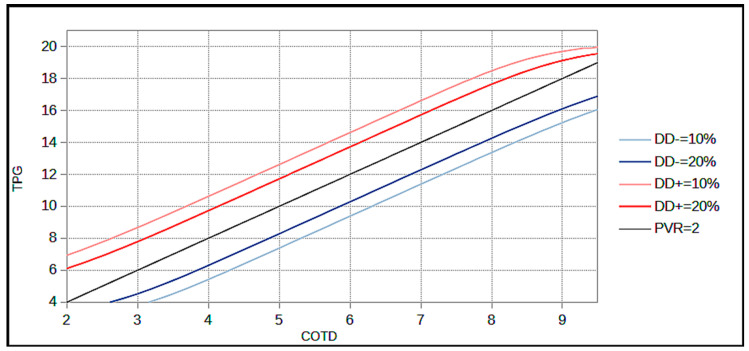
IsoDgDis (diagnosis disagreement) graph: IsoDgDis (DD) lines are represented on the TPG/CO_TD_ graph. The limits of the models are with physiological CO in between 1.3 L/min to 10.2 L/min, 1.96 SD of the bias of 2 L/min and bias of 0 L/min. The isoPVR line of 2 is the diagnosis line (patients above this line are diagnosed with precapillary PH and patients under this line are diagnosed with unclassified PH). COTD: Cardiac Output using Thermodilution; DgDis (DD): diagnosis disagreement (DgDis (DD)+ corresponds to false positive diagnosis using TD; DgDis (DD)− corresponds to false negative diagnosis using TD); PVR: pulmonary vascular resistance; TPG: transpulmonary gradient.

**Figure 4 jcm-12-00410-f004:**
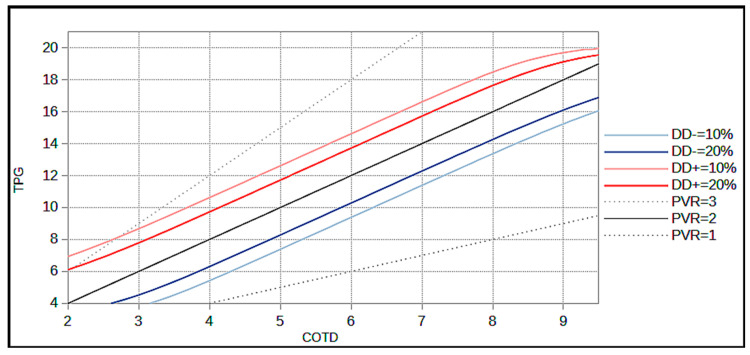
IsoDgDis and isoPVR graph. The same graph as above is shown with the addition of isoPVR line at 1 WU and 3 WU. This highlights the impact of CO and PVR values on diagnosis disagreement. COTD: Cardiac Output using Thermodilution; DgDis (DD): diagnosis disagreement (DgDis (DD)+) corresponds to false positive diagnosis using TD; DgDis (DD)− corresponds to false negative diagnosis using TD); PVR: pulmonary vascular resistance; TPG: transpulmonary gradient.

**Table 2 jcm-12-00410-t002:** Patients diagnosed with «unclassified PH» using TD and corresponding DgDis− probability.

TPG	CO_TD_	PVR_TD_	DgDis−
6	3.5	1.7	30.0%
7	4	1.75	29.4%
7	5	1.4	6.9%
8	5	1.6	16.2%
8	6	1.3	2.5%
9	5	1.8	31.1%
9	6	1.5	7.1%
10	6	1.7	15.4%
10	7	1.4	2.5%
11	6	1.8	31.2%
11	7	1.6	7.07%

DgDis− (diagnosis disagreement−) refers to the probability of being categorised as having PcPH using DF for a patient diagnosed with unclassified PH using TD (false negative). The TPG is equal to the difference between the mPAP and the PAWP. For example, a TPG of 11 corresponds to a mPAP of 21 and a PAWP of 10 or a mPAP of 22 and PAWP of 11. The percentage of DgDis− is higher when PVR is close to 2. In other words, the percentage of false negatives is much higher for the same TPG at lower CO_TD_ until it reaches a PVR of 2 WU. TPG: transpulmonary gradient; CO_TD_: Cardiac output measured by thermodilution; mPAP: mean pulmonary arterial pressure; PAWP: pulmonary artery wedge pressure; PH: pulmonary hypertension; PVR_TD_: pulmonary vascular resistance calculated using CO_TD_.

**Table 3 jcm-12-00410-t003:** Examples of patients diagnosed with PcPH using TD and corresponding DgDis+ probability.

TPG	CO_TD_	PVR_TD_	DgDis+
6	2.5	2.4	45.4%
7	2.5	2.8	23.8%
7	3	2.3	37.3%
8	3	2.7	19.6%
8	3.5	2.3	33.6%
9	3	3	8.5%
9	4	2.3	32.0%
10	3	3.3	3.0%
10	4	2.5	16.8%
11	4	2.75	7.3%
11	5	2.2	31.3%

DgDis+ (diagnosis disagreement+) refers to the probability of being categorised as having unclassified PH using DF for a patient diagnosed with PcPH using TD (false positive). The TPG is equal to the difference between the mPAP and the PAWP. For example, a TPG of 11 corresponds to a mPAP of 21 mmHg and a PAWP of 10 mmHg or a mPAP of 22 mmHg and PAWP of 11 mmHg. The percentage of DgDis+ is higher when PVR is close to 2. In other words, the percentage of false positives is higher for the same TPG at higher CO_TD_ until it reaches a PVR of 2 WU. TPG: transpulmonary gradient; CO_TD_: Cardiac output measured by thermodilution; mPAP: mean pulmonary arterial pressure; PAWP: pulmonary artery wedge pressure; PcPH: precapillary pulmonary hypertension; PVR: pulmonary vascular resistance.

## Data Availability

The data that support the findings of this study can be made available from the authors upon reasonable request.

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
