# Peer review of "The Influence of Methods for Cardiac Output Determination on the Diagnosis of Precapillary Pulmonary Hypertension: A Mathematical Model"

_jcm, 2023, doi:10.3390/jcm12020410_

Round 1

Reviewer 1 Report

With great interest I read the manuscript by Genecand et al. The strenght of this article is the exact determination of the reliability of different methods of cardiac output measurement. This has implications for daily practice: We must be aware that the classification of patients with low CO and low PVR needs cautious interpretation.

Minor remarks:

Study population:

As in the introduction, the study population should be characterized as the “...agreement between CO DF and CO TD in precapillary pulmonary hypertension”.

Discussion:

The method of indirect Fick (oxygen uptake from table of LaFarge) is less reliable, but the method itself should be mentioned in the discussion, for it is still used (mostly out of laziness) in clinical practice.

As remarked above (“study population”), the authors should add a comment on the methods of CO measurement outside of PAH / pcpPH, e.g. a study including 300 patients (Desole S et al. Comparison between thermodilution and Fick methods…Pulmonary Circulation 2022,12: e12128. DOI: 10.1002/pul2.12128) found a comparable superiority of COTD over CODF in patients with dyspnoea, insofar supporting the conclusion of the authors that COTD is the method in favour.

The references are exact, but could be a bit more extended. Maybe the authors consider some (not all) of the following publication on COTD over CODF in the discussion (here in alphabetical order):

- Espersen K, Jensen EW, Rosenborg D, Thomsen JK,Eliasen K, Olsen NV, Kanstrup IL. Comparison of cardiac output measurement techniques: thermodilution, Doppler, CO 2 ‐rebreathing and the direct Fick method. Acta Anaesthesiol Scand. 1995;39:245–51.

- Fares WH, Blanchard SK, Stouffer GA, Chang PP, Rosamond WD, Ford HJ, Aris RM. Thermodilution and Fick cardiac outputs differ: impact on pulmonary hypertension evaluation. Can Respir J. 2012;19:261–66.

- Grignola JC. Hemodynamic assessment of pulmonary hypertension. World J Cardiol. 2011;3:10–7.

- Kresoja KP, Faragli A, Abawi D, Paul O, Pieske B, Post H, Alogna A. Thermodilution vs estimated Fick cardiac output measurement in an elderly cohort of patients: a single‐centre experience. PLoS One. 2019;14:e0226561.

- Lipkin DP, Poole‐Wilson PA. Measurement of cardiac output during exercise by the thermodilution and direct Fick techniques in patients with chronic congestive heart failure. Am J Cardiol. 1985;56:321–24.

- Opotowsky AR, Hess E, Maron BA, Brittain EL, Barón AE, Maddox TM, Alshawabkeh LI, Wertheim BM, Xu M, Assad TR, Rich JD, Choudhary G, Tedford RJ. Thermodilution vs estimated Fick cardiac output measurement in clinical practice: an analysis of mortality from the Veterans Affairs Clinical Assessment, Reporting, and Tracking (VA CART) program and Vanderbilt University. JAMA Cardiol. 2017;2:1090–99.

- Raeside DA, Smith A, Brown A, Patel KR, Madhok R, Cleland J, Peacock AJ. Pulmonary artery pressure measurement during exercise testing in patients with suspected pulmonary hypertension. Eur Respir J. 2000;16:282–87.

Summary:

In the discussion the authors state exactly that “The DgDis is high if the patient lies close to a PVR of 2WU and the DgDis is higher for a given PVR for lower CO.” Maybe this sentence could be emphasized in a subsection as “take home massage”, if the publisher or the authors would like this form.

 I regret the short time frame of 7 days for the review. Nevertheless, the manuscript is very well prepared and nearly ready for publishing.

Author Response

We would like to thank the reviewer for the detailed reading of our work and precious comments which, we think, have improved the quality of our manuscript. Please find hereafter our answers.

The reviewer said: “As in the introduction, the study population should be characterized as the “...agreement between CO DF and CO TD in precapillary pulmonary hypertension”.

Response: “we have modified our manuscript accordingly and we also enriched our manuscript with two of your suggested references Fares et al and Desole et al and another two references from the literature concerning the precision of indirect Fick in Precapillary PH: Alkohdair et al and Rich et al”

The reviewer said: “The method of indirect Fick (oxygen uptake from table of LaFarge) is less reliable, but the method itself should be mentioned in the discussion, for it is still used (mostly out of laziness) in clinical practice.”

 Response: We have added a paragraph to the discussion session on this:

“The cardiac output measurement using indirect Fick (IF) is frequently used in clinic. However, the agreement between IF and TD for patients with PH or suspiscion of PH is poor with LoA ranging between 1.8L/min to 4.1 L/min and bias ranging from 0.4L/min to -0.8 L/min [5,12-14]. Comparisons of IF with DF in PcPH are lacking [7]. For this reason IF is not recommended anymore in the international guidelines because it is considered less reliable than TD [1]. Since the precision of the model was calibrated for TD it cannot be used for other methods, like IF, that have different precision.”

We also enriched the discussion on the use on non-invasive CO measurement in PcPH as well as the applicability of the model outside of PH:

“Other non-invasive methods have been studied in PcPH including inert gas rebreathing, bioimpendance, bioreactance, pulse wave analysis, cardiac magnetic resonnance imaging and echocardiography [7]. The most promising method might be cardiac magnetic resonnance imaging that showed excellent agreement when compared to TD or DF with LoA close to 1 L/min [15,16]. Since the characteristics of the presented mathematical model were based on the agreement between TD and DF, it cannot be readily used for other methods, like indirect Fick or cardiac magnetic resonnance, that have different agreement with the gold standard DF. However the mathematical model could be modified to study the influence on diagnosis disagreement of any given CO methods assuming that the LoA and bias are known and that the CO differences of the studied methods have a normal distribution.

The principle of this mathematical model could also be adapted to study the influence on the diagnosis of PcPH of different parameters of the PVR equation. For example the mPAP can be estimated using cardiac magnetic resonnance [17]. The mathematical model could be adapted to study the diagnosis influence of the mPAP estimated using cardiac magnetic resonnance compared to the mPAP measured during RHC.”

The reviewer said : “As remarked above (“study population”), the authors should add a comment on the methods of CO measurement outside of PAH / pcPH, e.g. a study including 300 patients (Desole S et al. Comparison between thermodilution and Fick methods…Pulmonary Circulation 2022,12: e12128.DOI: 10.1002/pul2.12128) found a comparable superiority of COTD over CODF in patients with dyspnoea, insofar supporting the conclusion of the authors that COTD is the method in favour.”

Response: we do not think that this article can lead to the conclusion that TD is superior to DF. Unexpectedly the agreement between TD and DF are wide (approximately 4L/min), even at rest, and similar to the agreement between TD and IF (approximately 4L/min) in this population. To the best of our knowledge, there is no data published so far demonstrating such a poor agreement between DF and TD. Also we never saw any article in the literature showing a precision similar between IF and DF when compared to TD. In this context, we wonder exactly how was performed DF. This is unfortunately not fully detailed in the article (calibration for V’O2 measurement? arterial sampling for CaO2 measurement or estimation using a pulse oximeter? mixed venous sampling in the PA concomitant to the arterial sampling?). Also the retrospective design might have biased the results. However, this study is very interesting and shows, once again, that the precision of the measurement of CO is subject to a lot of imprecisions, even with invasive measurements, and we have added this reference in our discussion concerning the imprecision of indirect Fick.

The reviewer said: “The references are exact, but could be a bit more extended. Maybe the authors consider some (not all) of the following publication on COTD over CODF in the discussion (here in alphabetical order):”

Response: we have added some of the suggested references.

The reviewer said: “in the discussion the authors state exactly that “The DgDis is high if the patient lies close to a PVR of 2WU and the DgDis is higher for a given PVR for lower CO.” Maybe this sentence could be emphasized in a subsection as “take home massage”, if the publisher or the authors would like this form.

Response: We thank the reviewer for this final point and we will ask JCM if it is possible to add a take home message.

Reviewer 2 Report

The influence of methods for cardiac output determination on the diagnosis of precapillary pulmonary hypertension: a mathematical model.

Overall, I would like to congratulate the authors on their endeavor. These mathematical assessments are important.

I will first go through line-by-line comments and then some general comments.

Line-by- Line:

Line 58 – widely replaced

Line 72 – (PcPH or unclassified PH)

Line 155 - PAWP, divided by CODF, equal to or less than 2 (TPG/CODF ≤2).

General Comments.

1.  The mathematical calculations are based on Table 1 studies. There are only 134 actual data sets and 21 of them are on people with CTEPH. This should be emphasized. I am also not sure if there is a difference between PcPH associated with CTEPH vs. no CTEPH. Further, although you mention it in Figure1,  I think you might also emphasize that all the data from the Bland Altman graph were generated based on the results of the three studies and not actual data.

2.  For Table 2., comment that the percentage of false negatives is much higher for the same TPG at lower COTD and for Table 3., the percentage of false positives is higher for the same TPG at higher COTD.

3.  I am not sure what figures 3 and 4 contribute to the manuscript and doesn’t show clearly what you are trying to convey, which is that the disagreement is relatively constant at most COTD. If there is an important reason for these graphs, please explain, otherwise remove them.

4. In your discussion, lines 244-248, you discuss and example. I would add that these values correspond to row 2 on table 2. In lines 249-253 you give another example where one could be confident in the diagnosis. However, this data is not in Table 3. I suggest you put in table 3 or use an example in row 8 of table 3 (10,3 and 3.3) where the false positive rate is only 3.0% which is reassuring.

5. I think in your limitations section you should also add something about the fact that at PAP close to 20 and/or low CO measurements (either Fick or Thermodilution), leading to a PVR close to 2, the variability in the measurements will lead to different diagnoses at different times and so clinicians should not consider a PVR of 2 a hard cut off between PcPH and unclassified PH. That is, the smaller the values for TPG and CO. the greater the effect of any variability in their measurements on PVR because the values are a ratio.  

Author Response

We would like to thank the reviewer for the detailed reading of our work and precious comments which, we think, have improved the quality of our manuscript. Please find hereafter our answers.

The reviewer said: “Line 58 – widely replaced, Line 72 – (PcPH or unclassified PH), Line 155 - PAWP, divided by CODF, equal to or less than 2 (TPG/CODF ≤2).”

Response: We have modified the text accordingly.

The reviewer said: “The mathematical calculations are based on Table 1 studies. There are only 134 actual data sets and 21 of them are on people with CTEPH. This should be emphasized.”

Response: We agree with this important comment. We have added the following sentence in the discussion: “However it must be acknowledged that there a paucity of data, with only three studies cumulating a total of 134 PcPH patients with 21 CTEPH and 113 PAH patients.”

The reviewer said: “I am also not sure if there is a difference between PcPH associated with CTEPH vs. no CTEPH.”

Response: This a pertinent comment. Unfortunately there is no such detailed in the presented studies.

The reviewer said: “Further, although you mention it in Figure1, I think you might also emphasize that all the data from the Bland Altman graph were generated based on the results of the three studies and not actual data.”

Response: we have emphasized this in the text adding “with fictive data” in the figure presentation.

The reviewer said: “For Table 2., comment that the percentage of false negatives is much higher for the same TPG at lower COTD and for Table 3., the percentage of false positives is higher for the same TPG at higher COTD.”

Response : we have added the following sentence in the figure 2 description : “The percentage of DgDis – is higher when PVR is close to 2. In other words, the percentage of false negatives is much higher for the same TPG at lower COTD until it reaches a PVR of 2 WU.” And the figure 3 description : “The percentage of DgDis + is higher when PVR is close to 2. In other words, the percentage of false positives is higher for the same TPG at higher COTD until it reaches a PVR of 2 WU.”

The reviewer said : “I am not sure what figures 3 and 4 contribute to the manuscript and doesn’t show clearly what you are trying to convey, which is that the disagreement is relatively constant at most COTD. If there is an important reason for these graphs, please explain, otherwise remove them.” 

Response: We think these graphs are of paramount importance. It allows the reader to simply put a patient with his hemodynamic constellation in the graph 3 and estimates his DgDis probability without any need to use a complicated formula. Furthermore graph 4 shows the influence of changing PVR on the DgDis. DgDis is higher for a given PVR for lower CO.

The reviewer said : “In your discussion, lines 244-248, you discuss and example. I would add that these values correspond to row 2 on table 2. In lines 249-253 you give another example where one could be confident in the diagnosis. However, this data is not in Table 3. I suggest you put in table 3 or use an example in row 8 of table 3 (10,3 and 3.3) where the false positive rate is only 3.0% which is reassuring.”

 Response: we have modified the text accordingly.

The reviewer said: “I think in your limitations section you should also add something about the fact that at PAP close to 20 and/or low CO measurements (either Fick or Thermodilution), leading to a PVR close to 2, the variability in the measurements will lead to different diagnoses at different times and so clinicians should not consider a PVR of 2 a hard cut off between PcPH and unclassified PH. That is, the smaller the values for TPG and CO. the greater the effect of any variability in their measurements on PVR because the values are a ratio.”  

Response: we thank the reviewer for this final comment. We have added a sentence concerning this in the discussion. “In other terms, since the PVR is a ratio between the TPG and the CO, the smaller the PTG and CO are in absolute terms, the greater is the impact of a given variability on the calculation of the PVR and thus the diagnosis disagreement.”